# Relevance Topic Model for Unstructured Social Group Activity Recognition

**Fang Zhao**       **Yongzhen Huang**       **Liang Wang**       **Tieniu Tan**

Center for Research on Intelligent Perception and Computing

Institute of Automation, Chinese Academy of Sciences

`{fang.zhao,yzhuang,wangliang,tnt}@nlpr.ia.ac.cn`

## Abstract

Unstructured social group activity recognition in web videos is a challenging task due to 1) the semantic gap between class labels and low-level visual features and 2) the lack of labeled training data. To tackle this problem, we propose a "relevance topic model" for jointly learning meaningful mid-level representations upon bag-of-words (BoW) video representations and a classifier with sparse weights. In our approach, sparse Bayesian learning is incorporated into an undirected topic model (i.e., Replicated Softmax) to discover topics which are relevant to video classes and suitable for prediction. Rectified linear units are utilized to increase the expressive power of topics so as to explain better video data containing complex contents and make variational inference tractable for the proposed model. An efficient variational EM algorithm is presented for model parameter estimation and inference. Experimental results on the Unstructured Social Activity Attribute dataset show that our model achieves state of the art performance and outperforms other supervised topic model in terms of classification accuracy, particularly in the case of a very small number of labeled training videos.

## 1   Introduction

The explosive growth of web videos makes automatic video classification important for online video search and indexing. Classifying short video clips which contain simple motions and actions has been solved well in standard datasets (such as KTH [1], UCF-Sports [2] and UCF50 [3]). However, detecting complex activities, specially social group activities [4], in web videos is a more difficult task because of unstructured activity context and complex multi-object interaction.

In this paper, we focus on the task of automatic classification of unstructured social group activities (e.g., wedding dance, birthday party and graduation ceremony in Figure 1), where the low-level features have innate limitations in semantic description of the underlying video data and only a few labeled training videos are available. Thus, a common method is to learn human-defined (or semi-human-defined) semantic concepts as mid-level representations to help video classification [4]. However, those human defined concepts are hardly generalized to a larger or new dataset. To discover more powerful representations for classification, we propose a novel supervised topic model called "relevance topic model" to automatically extract latent "relevance" topics from bag-of-words (BoW) video representations and simultaneously learn a classifier with sparse weights.

Our model is built on Replicated Softmax [5], an undirected topic model which can be viewed as a family of different-sized Restricted Boltzmann Machines that share parameters. Sparse Bayesian learning [6] is incorporated to guide the topic model towards learning more predictive topics which are associated with sparse classifier weights. We refer to those topics corresponding to non-zero weights as "relevance" topics. Meanwhile, binary stochastic units in Replicated Softmax are replaced by rectified linear units [7], which allows each unit to express more information for better

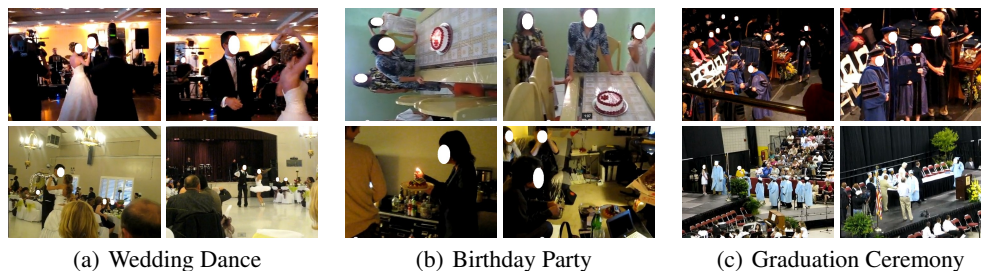

| (a) Wedding Dance | (b) Birthday Party | (c) Graduation Ceremony |

Figure 1: Example videos of the "Wedding Dance", "Birthday Party" and "Graduation Ceremony" classes taken from the USAA dataset [4].

explaining video data containing complex content and also makes variational inference tractable for the proposed model. Furthermore, by using a simple quadratic bound on the log-sum-exp function [8], an efficient variational EM algorithm is developed for parameter estimation and inference. Our model is able to be naturally extended to deal with multi-modal data without changing the learning and inference procedures, which is beneficial for video classification tasks.

## 2  Related work

The problems of activity analysis and recognition have been widely studied. However, most of the existing works [9, 10] were done on constrained videos with limited contents (e.g., clean background and little camera motions). Complex activity recognition in web videos, such as social group activity, is not much explored. Most relevant to our work is a recent work that learns video attributes to analyze social group activity [4]. In [4], a semi-latent attribute space is introduced, which consists of user-defined attributes, class-conditional and background latent attributes, and an extended Latent Dirichlet Allocation (LDA) [11] is used to model those attributes as topics. Different from that, our work discovers a set of discriminative latent topics without extra human annotations on videos.

From the view of graphical models, most similar to our model are the maximum entropy discrimination LDA (MedLDA) [12] and the generative Classification Restricted Boltzmann Machines (gClass-RBM) [13], both of which have been successfully applied to document semantic analysis. MedLDA integrates the max-margin learning and hierarchical directed topic models by optimizing a single objective function with a set of expected margin constraints. MedLDA tries to estimate parameters and find latent topics in a max-margin sense, which is different from our model relying on the principle of automatic relevance determination [14]. The gClassRBM used to model word count data is actually a supervised Replicated Softmax. Different from the gClassRBM, instead of point estimation of classifier parameters, our proposed model learns a sparse posterior distribution over parameters within a Bayesian paradigm.

## 3  Models and algorithms

We start with the description of Replicated Softmax, and then by integrating it with sparse Bayesian learning, propose the relevance topic model for videos. Finally, we develop an efficient variational algorithm for inference and parameter estimation.

### 3.1  Replicated Softmax

The Replicated Softmax model is a two-layer undirected graphical model, which can be used to model sparse word count data and extract latent semantic topics from document collections. Replicated Softmax allows for very efficient inference and learning, and outperforms LDA in terms of both the generalization performance and the retrieval accuracy on text datasets.

As shown in Figure 2 (left), this model is a generalization of the restricted Boltzmann machine (RBM). The bottom layer represents a multinomial visible unit sampled $K$ times ($K$ is the total number of words in a document) and the top layer represents binary stochastic hidden units.

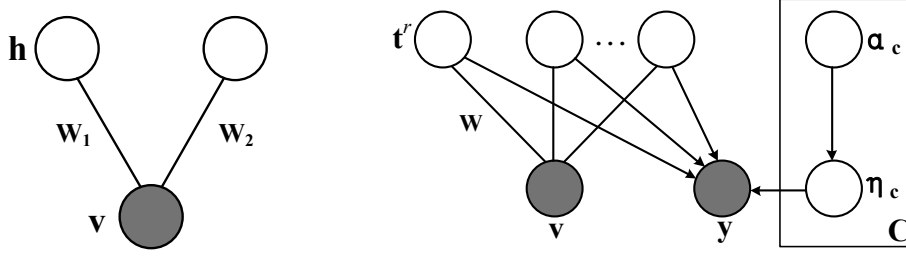

Figure 2: **Left:** Replicated Softmax model: an undirected graphical model. **Right:** Relevance topic model: a mixed graphical model. The undirected part models the marginal distribution of video BoW vectors $\mathbf{v}$ and the directed part models the conditional distribution of video classes $y$ given latent topics $\mathbf{t}^r$ by using a hierarchical prior on weights $\boldsymbol{\eta}$.

Let a word count vector $\mathbf{v} \in \mathbb{N}^N$ be the visible unit ($N$ is the size of the vocabulary), and a binary topic vector $\mathbf{h} \in \{0,1\}^F$ be the hidden units. Then the energy function of the state $\{\mathbf{v}, \mathbf{h}\}$ is defined as follows:

$$E(\mathbf{v}, \mathbf{h}; \theta) = -\sum_{i=1}^{N} \sum_{j=1}^{F} W_{ij} v_i h_j - \sum_{i=1}^{N} a_i v_i - K \sum_{j=1}^{F} b_j h_j, \tag{1}$$

where $\theta = \{\mathbf{W}, \mathbf{a}, \mathbf{b}\}$, $W_{ij}$ is the weight connected with $v_i$ and $h_j$, $a_i$ and $b_j$ are the bias terms of visible and hidden units respectively. The joint distribution over the visible and hidden units is defined by:

$$P(\mathbf{v}, \mathbf{h}; \theta) = \frac{1}{\mathcal{Z}(\theta)} \exp(-E(\mathbf{v}, \mathbf{h}; \theta)), \ \ \mathcal{Z}(\theta) = \sum_{\mathbf{v}} \sum_{\mathbf{h}} \exp(-E(\mathbf{v}, \mathbf{h}; \theta)), \tag{2}$$

where $\mathcal{Z}(\theta)$ is the partition function. Since exact maximum likelihood learning is intractable, the contrastive divergence [15] approximation is often used to estimate model parameters in practice.

## 3.2 Relevance topic model

The relevance topic model (RTM) is an integration of sparse Bayesian learning and Replicated Softmax, the main idea of which is to jointly learn discriminative topics as mid-level video representations and sparse discriminant function as a video classifier.

We represent the video dataset with class labels $y \in \{1, ..., C\}$ as $\mathcal{D} = \{(\mathbf{v}_m, y_m)\}_{m=1}^M$, where each video is represented as a BoW vector $\mathbf{v} \in \mathbb{N}^N$. Consider modeling video BoW vectors using the Replicated Softmax. Let $\mathbf{t}^r = [t_1^r, ..., t_F^r]$ denotes a $F$-dimensional topic vector of one video. According to Equation 2, the marginal distribution over the BoW vector $\mathbf{v}$ is given by:

$$P(\mathbf{v}; \theta) = \frac{1}{\mathcal{Z}(\theta)} \sum_{\mathbf{t}^r} \exp(-E(\mathbf{v}, \mathbf{t}^r; \theta)), \tag{3}$$

Since videos contain more complex and diverse contents than documents, binary topics are far from ideal to explain video data. We replace binary hidden units in the original Replicated Softmax with rectified linear units which are given by:

$$t_j^r = \max(0, t_j), \ P(t_j | \mathbf{v}; \theta) = \mathcal{N}(t_j | K b_j + \sum_{i=1}^{N} W_{ij} v_i, 1), \tag{4}$$

where $\mathcal{N}(\cdot | \mu, \tau)$ denotes a Gaussian distribution with mean $\mu$ and variance $\tau$. The rectified linear units taking nonnegative real values can preserve information about relative importance of topics. Meanwhile, the rectified Gaussian distribution is semi-conjugate to the Gaussian likelihood. This facilitates the development of variational algorithms for posterior inference and parameter estimation, which we describe in Section 3.3.

Let $\boldsymbol{\eta} = \{\boldsymbol{\eta}_y\}_{y=1}^C$ denote a set of class-specific weight vectors. We define the discriminant function as a linear combination of topics: $F(y, \mathbf{t}^r, \boldsymbol{\eta}) = \boldsymbol{\eta}_y^T \mathbf{t}^r$. The conditional distribution of classes is

defined as follows:

$$P(y|\mathbf{t}^r, \boldsymbol{\eta}) = \frac{\exp(F(y, \mathbf{t}^r, \boldsymbol{\eta}))}{\sum_{y'=1}^{C} \exp(F(y', \mathbf{t}^r, \boldsymbol{\eta}))}, \tag{5}$$

and the classifier is given by:

$$\hat{y} = \arg\max_{y \in C} \mathbb{E}[F(y, \mathbf{t}^r, \boldsymbol{\eta})|\mathbf{v}]. \tag{6}$$

The weights $\boldsymbol{\eta}$ are given a zero-mean Gaussian prior:

$$P(\boldsymbol{\eta}|\boldsymbol{\alpha}) = \prod_{y=1}^{C} \prod_{j=1}^{F} P(\eta_{yj}|\alpha_{yj}) = \prod_{y=1}^{C} \prod_{j=1}^{F} N(\eta_{yj}|0, \alpha_{yj}^{-1}), \tag{7}$$

where $\boldsymbol{\alpha} = \{\boldsymbol{\alpha}_y\}_{y=1}^{C}$ is a set of hyperparameter vectors, and each hyperparameter $\alpha_{yj}$ is assigned independently to each weight $\eta_{yj}$. The hyperpriors over $\boldsymbol{\alpha}$ are given by Gamma distributions:

$$P(\boldsymbol{\alpha}) = \prod_{y=1}^{C} \prod_{j=1}^{F} P(\alpha_{yj}) = \prod_{y=1}^{C} \prod_{j=1}^{F} \Gamma(c)^{-1} d^c \alpha_{yj}^{c-1} e^{-d\alpha}, \tag{8}$$

where $\Gamma(c)$ is the Gamma function. To obtain broad hyperpriors, we set $c$ and $d$ to small values, e.g., $c = d = 10^{-4}$. This hierarchical prior, which is a type of automatic relevance determination prior [14], enables the posterior probability of the weights $\boldsymbol{\eta}$ to be concentrated at zero and thus effectively switch off the corresponding topics that are considered to be irrelevant to classification.

Finally, given the parameters $\theta$, RTM defines the joint distribution:

$$P(\mathbf{v}, y, \mathbf{t}^r, \boldsymbol{\eta}, \boldsymbol{\alpha}; \theta) = P(\mathbf{v}; \theta)P(y|\mathbf{t}^r, \boldsymbol{\eta})\left(\prod_{j=1}^{F} P(t_j|\mathbf{v}; \theta)\right)\left(\prod_{y=1}^{C} \prod_{j=1}^{F} P(\eta_{yj}|\alpha_{yj})P(\alpha_{yj})\right). \tag{9}$$

Figure 2 (right) illustrates RTM as a mixed graphical model with undirected and directed edges. The undirected part models the marginal distribution of video data and the directed part models the conditional distribution of classes given latent topics. We can naturally extend RTM to **Multimodal RTM** by using the undirected part to model the multimodal data $\mathbf{v} = \{\mathbf{v}^{\mathrm{mod}l}\}_{l=1}^{L}$. Accordingly, $P(\mathbf{v}; \theta)$ in Equation 9 is replaced with $\prod_{l=1}^{L} P(\mathbf{v}^{\mathrm{mod}l}; \theta^{\mathrm{mod}l})$. In Section 3.3, we can see that it will not change learning and inference rules.

## 3.3 Parameter estimation and inference

For RTM, we wish to find parameters $\theta = \{\mathbf{W}, \mathbf{a}, \mathbf{b}\}$ that maximize the log likelihood on $\mathcal{D}$:

$$\log P(\mathcal{D}; \theta) = \log \int P(\{\mathbf{v}_m, y_m, \mathbf{t}_m^r\}_{m=1}^{M}, \boldsymbol{\eta}, \boldsymbol{\alpha}; \theta) d\{\mathbf{t}_m\}_{m=1}^{M} d\boldsymbol{\eta} d\boldsymbol{\alpha}, \tag{10}$$

and learn the posterior distribution $P(\boldsymbol{\eta}, \boldsymbol{\alpha}|\mathcal{D}; \theta) = P(\boldsymbol{\eta}, \boldsymbol{\alpha}, \mathcal{D}; \theta)/P(\mathcal{D}; \theta)$. Since exactly computing $P(\mathcal{D}; \theta)$ is intractable, we employ variational methods to optimize a lower bound $\mathcal{L}$ on the log likelihood by introducing a variational distribution to approximate $P(\{\mathbf{t}_m\}_{m=1}^{M}, \boldsymbol{\eta}, \boldsymbol{\alpha}|\mathcal{D}; \theta)$:

$$Q(\{\mathbf{t}_m\}_{m=1}^{M}, \boldsymbol{\eta}, \boldsymbol{\alpha}) = \left(\prod_{m=1}^{M} \prod_{j=1}^{F} q(t_{mj})\right) q(\boldsymbol{\eta})q(\boldsymbol{\alpha}). \tag{11}$$

Using Jensens inequality, we have:

$$\log P(\mathcal{D}; \theta) \geqslant \int Q(\{\mathbf{t}_m\}_{m=1}^{M}, \boldsymbol{\eta}, \boldsymbol{\alpha})$$

$$\log \frac{\left(\prod_{m=1}^{M} P(\mathbf{v}_m; \theta)P(y_m|\mathbf{t}_m^r, \boldsymbol{\eta})P(\mathbf{t}_m|\mathbf{v}_m; \theta)\right) P(\boldsymbol{\eta}|\boldsymbol{\alpha})P(\boldsymbol{\alpha})}{Q(\{\mathbf{t}_m\}_{m=1}^{M}, \boldsymbol{\eta}, \boldsymbol{\alpha})} d\{\mathbf{t}_m\}_{m=1}^{M} d\boldsymbol{\eta} d\boldsymbol{\alpha}. \tag{12}$$

Note that $P(y_m|\mathbf{t}_m^r, \boldsymbol{\eta})$ is not conjugate to the Gaussian prior, which makes it intractable to compute the variational factors $q(\boldsymbol{\eta})$ and $q(t_{mj})$. Here we use a quadratic bound on the log-sum-exp (LSE) function [8] to derive a further bound. We rewrite $P(y_m|\mathbf{t}_m^r, \boldsymbol{\eta})$ as follows:

$$P(y_m|\mathbf{t}_m^r, \boldsymbol{\eta}) = \exp(\mathbf{y}_m^T \mathbf{T}_m^r \boldsymbol{\eta} - \mathrm{lse}(\mathbf{T}_m^r \boldsymbol{\eta})), \tag{13}$$

where $\mathbf{T}_m^r \boldsymbol{\eta} = [(\mathbf{t}_m^r)^T \boldsymbol{\eta}_1, ..., (\mathbf{t}_m^r)^T \boldsymbol{\eta}_{C-1}]$, $\mathbf{y}_m = \mathbb{I}(y_m = c)$ is the one-of-C encoding of class label $y_m$ and $\mathrm{lse}(\mathbf{x}) \triangleq \log(1 + \sum_{y'=1}^{C-1} \exp(x_{y'}))$ (we set $\boldsymbol{\eta}_C = \mathbf{0}$ to ensure identifiability). In [8], the LSE function is expanded as a second order Taylor series around a point $\boldsymbol{\varphi}$, and an upper bound is found by replacing the Hessian matrix $\mathbf{H}(\boldsymbol{\varphi})$ with a fixed matrix $\mathbf{A} = \frac{1}{2}[\mathbf{I}_{C^*} - \frac{1}{C^*+1}\mathbf{1}_{C^*}\mathbf{1}_{C^*}^T]$ such that $\mathbf{A} \succ \mathbf{H}(\boldsymbol{\varphi})$, where $C^* = C - 1$, $\mathbf{I}_{C^*}$ is the identity matrix of size $M \times M$ and $\mathbf{1}_{C^*}$ is a $M$-vector of ones. Thus, similar to [16], we have:

$$\log P(y_m|\mathbf{t}_m^r, \boldsymbol{\eta}) \geqslant J(y_m, \mathbf{t}_m^r, \boldsymbol{\eta}, \boldsymbol{\varphi}_m) = \mathbf{y}_m^T \mathbf{T}_m^r \boldsymbol{\eta} - \frac{1}{2}(\mathbf{T}_m^r \boldsymbol{\eta})^T \mathbf{A} \mathbf{T}_m^r \boldsymbol{\eta} + \mathbf{s}_m^T \mathbf{T}_m^r \boldsymbol{\eta} - \kappa_i, \quad (14)$$

$$\mathbf{s}_m = \mathbf{A}\boldsymbol{\varphi}_m - \exp(\boldsymbol{\varphi}_m - \mathrm{lse}(\boldsymbol{\varphi}_m)), \quad (15)$$

$$\kappa_i = \frac{1}{2}\boldsymbol{\varphi}_m^T \mathbf{A}\boldsymbol{\varphi}_m - \boldsymbol{\varphi}_m^T \exp(\boldsymbol{\varphi}_m - \mathrm{lse}(\boldsymbol{\varphi}_m)) + \mathrm{lse}(\boldsymbol{\varphi}_m), \quad (16)$$

where $\boldsymbol{\varphi}_m \in \mathbb{R}^{C^*}$ is a vector of variational parameters. Substituting $J(y_m, \mathbf{t}_m^r, \boldsymbol{\eta}, \boldsymbol{\varphi}_m)$ into Equation 11, we can obtain a further lower bound:

$$\log P(\mathcal{D}; \theta) \geqslant \mathcal{L}(\theta, \boldsymbol{\varphi}) = \sum_{m=1}^{M} \log P(\mathbf{v}_m; \theta) + \mathbb{E}_Q\bigg[ \sum_{m=1}^{M} J(y_m, \mathbf{t}_m^r, \boldsymbol{\eta}, \boldsymbol{\varphi}_m)$$

$$+ \sum_{m=1}^{M} \log P(\mathbf{t}_m|\mathbf{v}_m; \theta) + \log P(\boldsymbol{\eta}|\boldsymbol{\alpha}) + \log P(\boldsymbol{\alpha}) - Q(\{\mathbf{t}_m\}_{m=1}^{M}, \boldsymbol{\eta}, \boldsymbol{\alpha}) \bigg]. \quad (17)$$

Now we convert the problem of model training into maximizing the lower bound $\mathcal{L}(\theta, \boldsymbol{\varphi})$ with respect to the variational posteriors $q(\boldsymbol{\eta})$, $q(\boldsymbol{\alpha})$ and $q(\mathbf{t}) = \{q(t_{mj})\}$ as well as the parameters $\theta$ and $\boldsymbol{\varphi} = \{\boldsymbol{\varphi}_m\}$. We can give some insights into the objective function $\mathcal{L}(\theta, \boldsymbol{\varphi})$: the first term is exactly the marginal log likelihood of video data and the second term is a variational bound of the conditional log likelihood of classes, thus maximizing $\mathcal{L}(\theta, \boldsymbol{\varphi})$ is equivalent to finding a set of model parameters and latent topics which could fit video data well and simultaneously make good predictions for video classes.

Due to the conjugacy properties of the chosen distributions, we can directly calculate free-form variational posteriors $q(\boldsymbol{\eta})$, $q(\boldsymbol{\alpha})$ and parameters $\boldsymbol{\varphi}$:

$$q(\boldsymbol{\eta}) = \mathcal{N}(\boldsymbol{\eta}|\mathbf{E}_{\boldsymbol{\eta}}, \mathbf{V}_{\boldsymbol{\eta}}), \quad (18)$$

$$q(\boldsymbol{\alpha}) = \prod_{y=1}^{C} \prod_{j=1}^{F} \mathrm{Gamma}(\alpha_{yj}|\hat{c}, \hat{d}_{yj}), \quad (19)$$

$$\boldsymbol{\varphi} = \langle \mathbf{T}_m^r \rangle_{q(\mathbf{t})} \mathbf{E}_{\boldsymbol{\eta}}, \quad (20)$$

where $\langle \cdot \rangle_q$ denotes an exception with respect to the distribution $q$ and

$$\mathbf{V}_{\boldsymbol{\eta}} = \bigg( \sum_{m=1}^{M} \langle (\mathbf{T}_m^r)^T \mathbf{A} \mathbf{T}_m^r \rangle_{q(\mathbf{t})} + \mathrm{diag}\langle \alpha_{yj} \rangle_{q(\boldsymbol{\alpha})} \bigg)^{-1}, \mathbf{E}_{\boldsymbol{\eta}} = \mathbf{V}_{\boldsymbol{\eta}} \sum_{m=1}^{M} \langle (\mathbf{T}_m^r)^T \rangle_{q(\mathbf{t})} (\mathbf{y}_m + \mathbf{s}_m), \quad (21)$$

$$\hat{c} = c + \frac{1}{2}, \hat{d}_{yj} = d + \frac{1}{2}\langle \eta_{yj}^2 \rangle_{q(\boldsymbol{\eta})}. \quad (22)$$

For $q(\mathbf{t})$, the calculation is not immediate because of the rectification. Inspired by [17], we have the following free-form solution:

$$q(t_{mj}) = \frac{\omega_{pos}}{Z}\mathcal{N}(t_{mj}|\mu_{pos}, \sigma_{pos}^2)u(t_{mj}) + \frac{\omega_{neg}}{Z}\mathcal{N}(t_{mj}|\mu_{neg}, \sigma_{neg}^2)u(-t_{mj}), \quad (23)$$

where $u(\cdot)$ is the unit step function. See Appendix A for parameters of $q(t_{mj})$.

Given $\theta$, through repeating the updates of Equations 18-20 and 23 to maximize $\mathcal{L}(\theta, \boldsymbol{\varphi})$, we can obtain the variational posteriors $q(\boldsymbol{\eta})$, $q(\boldsymbol{\alpha})$ and $q(\mathbf{t})$. Then given $q(\boldsymbol{\eta})$, $q(\boldsymbol{\alpha})$ and $q(\mathbf{t})$, we estimate $\theta$ by using stochastic gradient descent to maximize $\mathcal{L}(\theta, \boldsymbol{\varphi})$, and the derivatives of $\mathcal{L}(\theta, \boldsymbol{\varphi})$ with

respect to $\theta$ are given by:

$$\frac{\partial \mathcal{L}(\theta, \boldsymbol{\varphi})}{\partial W_{ij}} = \langle v_i t_j^r \rangle_{data} - \langle v_i t_j^r \rangle_{model} + \frac{1}{M} \sum_{m=1}^{M} v_{mi} \left( \langle t_{mj} \rangle_{q(\mathbf{t})} - \sum_{i=1}^{N} W_{ij} v_{mi} - K b_j \right), \quad (24)$$

$$\frac{\partial \mathcal{L}(\theta, \boldsymbol{\varphi})}{\partial a_i} = \langle v_i \rangle_{data} - \langle v_i \rangle_{model}, \quad (25)$$

$$\frac{\partial \mathcal{L}(\theta, \boldsymbol{\varphi})}{\partial b_j} = \langle t_j^r \rangle_{data} - \langle t_j^r \rangle_{model} + \frac{K}{M} \sum_{m=1}^{M} \left( \langle t_{mj} \rangle_{q(\mathbf{t})} - \sum_{i=1}^{N} W_{ij} v_{mi} - K b_j \right), \quad (26)$$

where the derivatives of $\sum_{m=1}^{M} \log P(\mathbf{v}_m; \theta)$ are the same as those in [5].

This leads to the following variational EM algorithm:

**E-step:** Calculate variational posteriors $q(\boldsymbol{\eta})$, $q(\boldsymbol{\alpha})$ and $q(\mathbf{t})$.
**M-step:** Estimate parameters $\theta = \{\mathbf{W}, \mathbf{a}, \mathbf{b}\}$ through maximizing $\mathcal{L}(\theta, \boldsymbol{\varphi})$.

These two steps are repeated until $\mathcal{L}(\theta, \boldsymbol{\varphi})$ converges. For the Multimodal RTM learning, we just additionally calculate the gradients of $\theta^{\text{mod}l}$ for each modality $l$ in the M-step while the updating rules are not changed.

After the learning is completed, according to Equation 6 the prediction for new videos can be easily obtained:

$$\hat{y} = \arg \max_{y \in C} \left\langle \boldsymbol{\eta}_y^T \right\rangle_{q(\boldsymbol{\eta})} \left\langle \mathbf{t}^r \right\rangle_{p(\mathbf{t}|\mathbf{v};\theta)}. \quad (27)$$

## 4 Experiments

We test our models on the Unstructured Social Activity Attribute (USAA) dataset [1] for social group activity recognition. Firstly, we present quantitative evaluations of RTM in the case of different modalities and comparisons with other supervised topic models (namely MedLDA and gClass-RBM). Secondly, we compare Multimodal RTM with some baselines in the case of plentiful and sparse training data respectively. In all experiments, the contrastive divergence is used to efficiently approximate the derivatives of the marginal log likelihood and the unsupervised training on Replicated Softmax is used to initialize $\theta$.

### 4.1 Dataset and video representation

The USAA dataset consists of 8 semantic classes of social activity videos collected from the Internet. The eight classes are: birthday party, graduation party, music performance, non-music performance, parade, wedding ceremony, wedding dance and wedding reception. The dataset contains a total of 1466 videos and approximate 100 videos per-class for training and testing respectively. These videos range from 20 seconds to 8 minutes averaging 3 minutes and contain very complex and diverse contents, which brings significant challenges for content analysis.

Each video is represented using three modalities, i.e., static appearance, motion, and auditory. Specifically, three visual and audio local keypoint features are extracted for each video: scale-invariant feature transform (SIFT) [18], spatial-temporal interest points (STIP) [19] and mel-frequency cepstral coefficients (MFCC) [20]. Then the three features are collected into a BoW vector (5000 dimensions for SIFT and STIP, and 4000 dimensions for MFCC) using a soft-weighting clustering algorithm, respectively.

### 4.2 Model comparisons

To evaluate the discriminative power of video topics learned by RTM, we present quantitative classification results compared with other supervised topic models (MedLDA and gClassRBM) in the case of different modalities. We have tried our best to tune these compared models and report the best results.

Table 1: Classification accuracy of different supervised topic models for single-modal features.

| Feature | | SIFT | | | STIP | | | MFCC | | |
|---------|---|---|---|---|---|---|---|---|---|---|
| Model | | Med LDA | gClass RBM | RTM | Med LDA | gClass RBM | RTM | Med LDA | gClass RBM | RTM |
| Accuracy (%) | 20 topics | 44.72 | 45.40 | **51.99** | 37.28 | 42.39 | **48.29** | 34.71 | 41.70 | **45.35** |
| | 30 topics | 44.17 | 46.11 | **53.09** | 38.93 | 42.25 | **49.11** | 38.55 | 43.62 | **46.67** |
| | 40 topics | 43.07 | 47.08 | **55.83** | 40.85 | 42.39 | **50.62** | 41.15 | 45.00 | **48.15** |
| | 50 topics | 42.80 | 46.81 | **54.17** | 39.75 | 41.70 | **51.71** | 41.98 | 44.31 | **47.46** |
| | 60 topics | 40.74 | 49.72 | **54.03** | 41.54 | 43.35 | **51.17** | 38.27 | 43.48 | **47.33** |

Table 2: Classification accuracy of different methods for multimodal features.

| Method | | Multimodal RTM | | RS+SVM | Direct | SVM-UD+LR | SLAS+LR |
|--------|---|---|---|---|---|---|---|
| Accuracy (%) | 100 Inst | 60 topics | 60.22 | 54.60 | | | |
| | | 90 topics | 62.69 | 56.10 | | | |
| | | 120 topics | 63.79 | 57.34 | **66.0** | 65.0 | 65.0 |
| | | 150 topics | 64.06 | 59.26 | | | |
| | | 180 topics | 64.72 | 60.63 | | | |
| | 10 Inst | 60 topics | 38.68 | 23.73 | | | |
| | | 90 topics | **41.29** | 28.53 | | | |
| | | 120 topics | **43.48** | 30.59 | 29.0 | 37.0 | 40.0 |
| | | 150 topics | **43.72** | 33.47 | | | |
| | | 180 topics | **44.99** | 35.94 | | | |

Table 1 shows the classification accuracy of different models for three single-modal features: SIFT, STIP and MFCC. We can see that RTM achieves higher classification accuracy than MedLDA and gClassRBM in all cases, which demonstrates that through leveraging sparse Bayesian learning to incorporate class label information into topic modeling, RTM can find more discriminative topical representations for complex video data.

## 4.3 Baseline comparisons

We compare Multimodal RTM with the baselines in [4] which are the best results on the USAA dataset:

**Direct** Direct SVM or KNN classification on raw video BoW vectors (14000 dimensions), where SVM is used for experiments with more than 10 instances and KNN otherwise.

**SVM-UD+LR** SVM attribute classifiers learn 69 user-defined attributes, and then a logistic regression (LR) classifier is performed according to the attribute classifier outputs.

**SLAS+LR** Semi-latent attribute space is learned, and then a LR classifier is performed based on the 69 user-defined, 8 class-conditional and 8 latent topics.

Besides, we also perform a comparison with another baseline where different modal topics extracted by Replicated Softmax are connected together as video representations, and then a multi-class SVM classifier [21] is learned from the representations. This baseline is denoted by RS+SVM.

The results are illustrated in Table 2. Here the number of topics of each modality is assumed to be the same. When the labeled training data is plentiful (100 instances per class), the classification performance of Multimodal RTM is similar to the baselines in [4]. However, We argue that our model learns a lower dimensional latent semantic space which provides efficient video representations and is able to be better generalized to a larger or new dataset because extra human defined concepts are not required in our model. When considering the classification scenario where only a very small number of training data are available (10 instances per class), Multimodal RTM can achieve better performance with an appropriate number (e.g., $\geqslant 90$) of topics because the sparsity of relevance topics learned by RTM can effectively prevent overfitting to specific training instances. In addition, our model outperforms RS+SVM in both cases, which demonstrates the advantage of jointly learning topics and classifier weights through sparse Bayesian learning.

It is also interesting to examine the sparsity of relevance topics. Figure 3 illustrates the degree of correlation between topics and two different classes. We can see that the learned relevance topics are very sparse, which leads to good generalisation for new instances and robustness for small datasets.

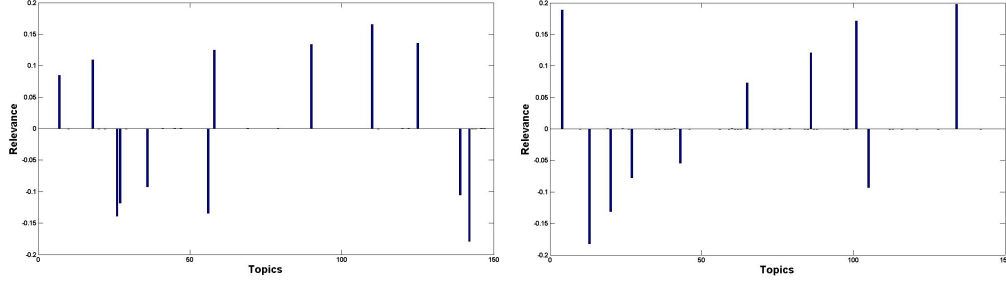

Figure 3: Relevance topics discovered by RTM for two different classes. Vertical axis indicates the degree of positive and negative correlation.

## 5    Conclusion

This paper has proposed a supervised topic model, the relevance topic model (RTM), to jointly learn discriminative latent topical representations and a sparse classifer for recognizing unstructured social group activity. In RTM, sparse Bayisian learning is integrated with an undirected topic model to discover sparse relevance topics. Rectified linear units are employed to better fit complex video data and facilitate the learning of the model. Efficient variational methods are developed for parameter estimation and inference. To further improve video classification performance, RTM is also extended to deal with multimodal data. Experimental results demonstrate that RTM can find more predictive video topics than other supervised topic models and achieve state of the art classification performance, particularly in the scenario of lacking labeled training videos.

## Appendix A. Parameters of free-form variational posterior $q(t_{mj})$

The expressions of parameters in $q(t_{mj})$ (Equation 23) are listed as follows:

$$\omega_{pos} = \mathcal{N}(\alpha|\beta, \gamma + 1), \ \sigma_{pos}^2 = (\gamma^{-1} + 1)^{-1}, \ \mu_{pos} = \sigma_{pos}^2 (\frac{\alpha}{\gamma} + \beta), \tag{28}$$

$$\omega_{neg} = \mathcal{N}(\alpha|0, \gamma), \ \sigma_{neg}^2 = 1, \ \mu_{neg} = \beta, \tag{29}$$

$$Z = \frac{1}{2}\omega_{pos}\text{erfc}\left(\frac{-\mu_{pos}}{\sqrt{2\sigma_{pos}^2}}\right) + \frac{1}{2}\omega_{neg}\text{erfc}\left(\frac{\mu_{neg}}{\sqrt{2\sigma_{neg}^2}}\right), \tag{30}$$

where $\text{erfc}(\cdot)$ is the complementary error function and

$$\alpha = \left\langle \frac{\boldsymbol{\eta}_{\cdot j}\left(\mathbf{y}_m + \mathbf{s}_m - \sum_{j'\neq j}\boldsymbol{\eta}_{\cdot j'}\mathbf{A}t_{mj'}^r\right)}{\boldsymbol{\eta}_{\cdot j}\mathbf{A}\boldsymbol{\eta}_{\cdot j}^T} \right\rangle_{q(\boldsymbol{\eta})q(\mathbf{t})}, \tag{31}$$

$$\gamma = \left\langle \boldsymbol{\eta}_{\cdot j}\mathbf{A}\boldsymbol{\eta}_{\cdot j}^T \right\rangle_{q(\boldsymbol{\eta})}^{-1}, \ \beta = \sum_{i=1}^{N} W_{ij}v_{mi} + Kb_j. \tag{32}$$

We can see that $q(t_{mj})$ depends on expectations over $\boldsymbol{\eta}$ and $\{t_{mj'}\}_{j'\neq j}$, which is consistent with the graphical model representation of RTM in Figure 2.

### Acknowledgments

This work was supported by the National Basic Research Program of China (2012CB316300), Hundred Talents Program of CAS, National Natural Science Foundation of China (61175003, 61135002, 61203252), and Tsinghua National Laboratory for Information Science and Technology Cross-discipline Foundation.

## Footnotes

[1]Available at `http://www.eecs.qmul.ac.uk/~yf300/USAA/download/`.

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
