[Reviews · NeurIPS 2013]

Submitted by Assigned_Reviewer_2

The authors build on the replicated softmax model to devise a classification approach that lends itself to the video classification with only a few examples. The model consists of two parts: The first part is similar to a replicated softmax model that captures topics in the features, with the difference of using a rectified linear units. The second part consists of a discriminant function as a linear combination of topics. The authors apply a trick to make inference tractable: they derive a variational bound which is further lower bounded exploiting the semi-conjugacy between the rectified linear units and the Gaussian likelihood. The model is evaluated on a dataset of videos of social activities, which are to be classified.

The problem is important, the model derivation is sound, the experimental evaluation is correct.

However, the approach is independent (and not motivated) from the application of unstructured social group activity recognition. It seems to me that the author developed a general feature learning and classification approach that should be evaluated against other classification datasets and further baselines.

Summary: An ok approach, but the approach has not much to do with social group activity recognition. As a generic feature learning and classification approach, many comparison to other base lines would strengthen the work.

Submitted by Assigned_Reviewer_4

 This paper considers automatic classification of unstructured social group activity videos. To bridge the semantic gap between low-level features and the class-labels, the authors adopt a latent topic model based on replicated softmax to extract topics as mid-level representations for video classification. The main idea of this paper is the integration of sparse Bayesian learning and replicated softmax, which leads to the proposed model referred to “relevance topic model (RTM)”. In RTM, the discriminative topics and sparse classifier weights are learned jointly, and the authors proposes variational EM algorithm for model parameter estimation and inference. The authors test their algorithm on a benchmark dataset and demonstrate better performance compared to other supervised topic models and some baseline algorithms.
 The paper seems to be well organized. It is well motivated and proposes ideas that are useful in relative area (e.g. video scene analysis, classification and recognition ). It cites relevant research papers adequately.
 It is difficult to understand the significance in using sparsity on the BOW representation, classifier and hierarchical prior on the weight. Also, it is not clear what the significant differences are between the attributes used in the previous research [4] and the topics discovered by the proposed RTM. It should be noted that the idea of using Replicated softmax model for extracting latent topics as mid-level representations is not novel [5]. However, the key idea of this paper is the joint learning of latent topics and classifier weights, which is interesting and novel.
 Section 3.3 contains many complex mathematical equations, which is not easy to follow.
 Experimental results demonstrate good performance of proposed algorithms. However, the experimental backup is rather weak due to the following two issues; 1) The performance in terms of generalization is not good. In case of using many instances (e.g 100 in Table2), the proposed algorithm could not achieve good results. 2) In Section 4.3, the authors compare the proposed algorithm with only the baselines in [4]. Since the novelty of the proposed algorithm in relation to the previous Replicated softmax model seems to be the joint learning of discriminative topics and sparse classifier weights, the reviewer suggest using other baselines where the topics are extracted via previous replicated softmax, but classifier weights are learned separately from the topics (e.g. latent topic extraction using replicated softmax[5] + SVM classifier).
Summary: The key idea of this paper is the joint learning of latent topics and classifier weights, which is interesting and will be useful in many related research fields (e.g. video context classification and recognition). In the experiment section, it is recommended to include additional baseline methods to compare to reveal the strength of the above key idea.

Submitted by Assigned_Reviewer_5

General:
The paper proposes a supervised (hybrid) topic model for unstructured activity recognition. The model is named: relevance topic model (RTM). The supervised part is the label of the classes of the training videos.

RTM is an integration of sparse Bayesian learning and Replicated Softmax. The main concept is to jointly learn discriminative topics as mid-level video representations and discriminant function as a video classifier. RTM is composed of an undirected part to model the marginal distribution of video
words and a directed part to model the conditional distribution of video classes given the latent topics. Also, the authors propose the parameter estimation and inference methods.
The authors evaluate their algorithm in the Unstructured Social Activity Attribute (USAA) presenting quantitative results of RTM. They compare the method with the literature improving in all cases the activity classification accuracy. Also the authors show an interesting comparison of the correlation of topics of two different classes.

The idea seems interesting. The usage of SIFT, STIP and MFCC which are very low level features mostly based on pixel representations. It can be interesting a small discussion on how more descriptive features (object detection) can be added to the model.

Relevant work that could be added in the Bibliography, [1] Similar subject [2] interesting approach on structured activity discovery.

[1] Social Role Discovery in Human Events
Vignesh Ramananthan, Bangpeng Yao, and Li Fei-Fei
IEEE Conference on Computer Vision and Pattern Recognition (CVPR). Portland, OR, USA. June 23-28, 2013
[2] J. Varadarajan, R. Emonet and J.-M. Odobez
Int. Journal of Computer Vision (IJCV), Vol. 103, Num. 1, pages 100-126, May 2013.

Quality:
The quality of the paper is good.

Clarity:
The paper is clear, the problem is well defined.

Originality:
The originality is Medium. This is not the first approach that aims at building intermediate features for activity recognition. Nor the first one that uses some full or semi-supervision.

Unsupervised methods such as [2] for structured scenarios have strong assumptions in the temporal structure of the words (observation vectors) but in unstructured activities as the ones target in this work temporal constraints can be relaxed.

Minimal supervision in the learning is acceptable for a difficult task as activity recognition in wild videos. An interesting question would be how many is the minimal human labeled data that is required to classify wild videos decently.

Significance:
The topic is exciting research, I would say of big significance.
Summary: The paper addresses an interesting topic, which is the recognition of social activities in unconstrained videos. The authors propose an interesting model, which uses minimal labeling for learning and discovering topics used for activity recognition. It is well written and with enough evaluation.
Author Feedback

Author rebuttal: We thank all the reviewers for their comments and suggestions. Especially, they consider that we proposed a novel and effective model (relevance topic model) to address the challenging problem in unstructured social group activity recognition by jointly learning discriminative latent topics and a classifier with sparse weights, with a small number of labeled training instances. We appreciate their positive assessments to our work, e.g., "idea is interesting and novel" (R4&R5), "derivation is sound" (R2), "evaluation is correct (R2), enough (R5), good performance (R4)", "paper is clear, and well organized and written" (R4&R5), and "of big significance" (R5).

The following responds to each of main concerns raised by reviewers.

To R2:

1. The approach has not much to do with social group activity recognition.

>> The proposed RTM is greatly motivated by two main challenging problems in unstructured social group activity recognition: 1) the semantic gap between low-level visual features and class labels and 2) the lack of labeled training data.

>> To address these two problems, for the former, RTM jointly learns latent relevance topics as mid-level representations and a classifier mapping the mid-level representations to class labels. For the letter, RTM forces the sparsity on relevance topics by sparse Bayesian learning to prevent overfitting to specific training instances, which is clearly validated in Section 4.3.

2. As a generic feature learning and classification approach, other base lines and experiments on other datasets would be necessary.

>> Our model mainly focuses on the task of unstructured social group activity recognition. We will try to generalize our model to other potential applications in future work.

To R4:

1. Significance of using sparsity.

>> Using sparsity is to make the relevance topics learned by RTM more interpretable and discriminative. It provides good generalization and robustness in the case of a small number of labeled training instances, which is experimentally justified in Section 4.3.

2. Difference between the attributes used in [4] and the topics discovered by RTM.

>> [4] models attributes as topics by LDA. It learns a SEMI-latent topic space because a part of topics are USER-DEFINED. The proposed RTM learns a COMPLETELY latent topic space, which can be easily generalized to any larger or new datasets.

3. In the case of using many instances, the performance is not good.

>> When using many (100) instances, our method is only slightly worse (1.28%) than [4], which is state-of-the-art, but better (4.99%) in the case of 10 instances. In addition to accuracy, our method learns a lower dimensional latent semantic space, which provides much more efficient representations.

4. Compare with additional baseline methods where the topics and the classifier are learned separately, e.g., replicated softmax + SVM.

>> Thanks for your suggestion, and we may include such experimental results. However, it should be noted that learning topics and a classifier separately is generally worse than learning jointly. A theoretical insight is that topic extraction is unsupervised in separate learning but supervised in joint learning, thus topics extracted in the latter are more predictive than those in the former. Besides, a similar case, i.e., LDA + SVM, is worse than joint learning, which has been demonstrated in [11].

To R5:

1. An interesting question would be how many is the minimal human labeled data that is required.

>> In our current experiment, 10 would be the minimal acceptable number for the used dataset. We will make an in-depth study on this problem in future work.

2. Relevant work that could be added in the Bibliography, [1] Similar subject [2] interesting approach on structured activity discovery.

>> Thanks for reminding these two fairly new papers (published after the submission of this paper). We will cite them.

Thank all reviewers again. We would also like to address several minor concerns, e.g., clarifying complex mathematical equations, if the paper is accepted.